# Multi-Agent Trajectory Prediction by Combining Egocentric and Allocentric Views

Xiaosong Jia[1], Liting Sun[2], Hang Zhao[3], Masayoshi Tomizuka[2], and Wei Zhan[2]

[1]Shanghai Jiao Tong University - jiaxiaosong1997@gmail.com
[2]University of California, Berkeley - {litingsun,wzhan}@berkeley.edu, tomizuka@me.berkeley.edu
[3]Tsinghua University - zhaohang0124@gmail.com

**Abstract:** Trajectory prediction of road participants such as vehicles and pedestrians is crucial for autonomous driving. Recently, graph neural network (GNN) is widely adopted to capture the social interactions among the agents. Many GNN-based models formulate the prediction task as a single-agent prediction problem where multiple inference is needed for multi-agent prediction (which is common in practice), which leads to fundamental inconsistency in terms of homotopy as well as inefficiency for the memory and time. Moreover, even for models that do perform joint prediction, typically one centric agent is selected and all other agents' information is normalized based on that. Such centric-only normalization leads to asymmetric encoding of different agents in GNN, which might harm its performance. In this work, we propose a efficient multi-agent prediction framework that can predict all agents' trajectories jointly by normalizing and processing all agents' information symmetrically and homogeneously with combined egocentirc and allocentric views. Experiments are conducted on two interaction-rich behavior datasets: INTERACTION (vehicles) and TrajNet++ (pedestrian). The results show that the proposed framework can significantly boost the inference speed of the GNN-based model for multi-agent prediction and achieve better performance. In the INTERACTION dataset's challenge, the proposed model achieved the 1st place in the regular track and generalization track.

**Keywords:** Autonomous Driving, Joint Trajectory Prediction, Multi-Agent Interaction

## 1 Introduction

Trajectory prediction of road participants like vehicles and pedestrians is of great significance for planning and decision making of autonomous vehicles. Recently, lots of research efforts have been devoted to capture the social interactions among the agents with feature pooling [1, 2] or graph neural network [3, 4, 5]. However, most of the state-of-the-art (SOTA) models [5, 6, 7] formulate the problem as single-agent prediction, i.e., the models try to learn the marginal distribution of what a target agent may act in the future given the historical observations of all agents' trajectories in the scene. In cases of multiple agent prediction, a straightforward way is to run inference multiple times [8], that is, onece for each agent. Under such circumstances, the "joint" prediction results might not be consistent in terms of homotopy, not to mention the low efficiency in terms of memory and time. There are also models which perform joint prediction for multiple agents [9, 10]. However, such work adopts a uniform coordinate for all modeled agents, i.e., a centric agent is selected and all coordinates are normalized based on the centric agent. Therefore, only the selected agent is

5th Conference on Robot Learning (CoRL 2021), London, UK.

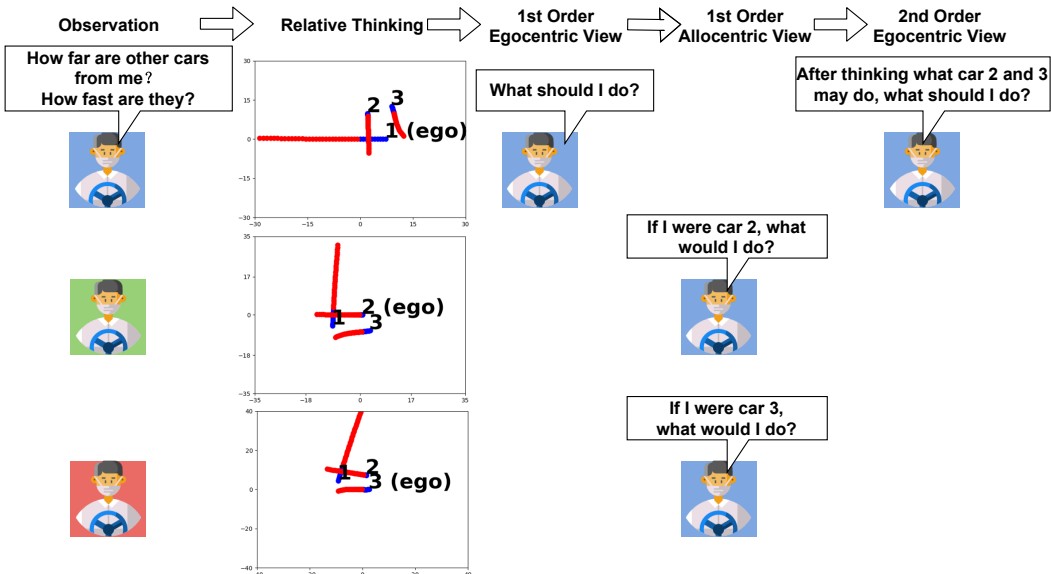

Figure 1: Illustration of the intuition for each stage of the proposed model - the thinking process of interacting agents. Here, we only show the first agent's (blue) thinking process since the other two are symmetrical. First, a human driver understands the world by estimating the relative distance and speed of other agents (Relative Thinking), which corresponds to the normalization stage of the proposed model. Then each agent considers how it should react (Egocentric View), which corresponds to the stage of updating the node features by aggregating neighbors' information in the proposed model. After that, an agent put itself in someone else's shoes for all neighbors (Allocentric View), which corresponds to the update of edge features by accessing source node's information in the proposed model. After alternatively repeating the Egocentric View stage and Allocentric View stage for several rounds, all agents make the decision according to the interacting situations.

providing egocentric views, while all other agents are in allocentric views.[1] The disadvantage of all data in a single egocentric view is two-fold. First, not all agents are treated in an unbiased way due to the lack of egocentric views from other agents, like missing the view of "putting in someone else's shoes". Second, the egocentric view of one agent makes the spatial relations between two other agents heterogeneous from the spatial relations between ego agent and other agents. The different distributions of input node feature might harm the GNN-based models' performance since it usually shares parameter for the aggregation and fusion function for all nodes.

In this work, we aim to design an efficient GNN-based multi-agent joint prediction model. Our key insight is that all agents should be treated symmetrically in the graph, providing both egocentric and allocentric views. Moreover, we decouple the update of the node and edge feature to improve the efficiency of the network. As a result, we propose to use the "Node update" step to collect and summarize other agents' information in a egocentric view, and then the "Edge update" step to collect information in allocentric views from each neighbor. Via such a combined egocentric-and-allocentric views, it is guaranteed that the graph model handles all agents homogeneously which will encourage the shared parameters to learn to extract the underlying interactions among agents. We illustrate the intuition behind the proposed model in Fig. 1

We conducted experiments on two interaction-rich datasets: INTERACTION (vehicles) [12] and TrajNet++ (pedestrian) [13]. The results show that compared to running multiple single prediction models, the efficiency is boosted 1.7-4.5 times higher while achieving better accuracy. In the IN-

---

[1]Egocentric and allocentric are terms usually used in the psychology and cognition area. According to [11], a study on human spatial cognition and animal navigation, they define the egocentric spatial coding system as the location of objects in space relative to the ego body and the allocentric spatial coding system as the location of one object is defined with respect to other objects. In this paper, we refer this two concepts. As for the coordinate system transformation, we set the reference's coordinate as the origin and yaw angle aligned with the negative x-axis.

TERACTION dataset's challenge, the model achieved the $1^{st}$ **place** in the regular track and generalization track [2] We have also conducted experiments on three efficient variants of the multi-inference single-prediction model under different view settings and the results show that the proposed model outperforms those variants, which demonstrates the advantage of combining the egocentric and allocentric views.

## 2 Related Works

Plenty of works have been done to capture the complex social interactions among multiple agents. There are pooling-based methods [1, 2] which use the mean/max/sum of all agents' hidden vectors to obtain a single vector to represent the social context. Social LSTM [1] utilized the social context of each time step as a part of the input for its next time step to the LSTM. Social GAN [2] obtained the social context by the pooling of output from the encoder RNN at the last time-step and fed the social context with each agents' hidden vector to the decoder to generate their future trajectories. Though pooling is fast, it loses too much information and lacks of expressive ability to output a specific context for each agent. There are also methods which renders the entire scene including the agents and HD Map onto images and captures the interactions implicitly by CNN. Hong et al. [14] adopted pure CNN to process the scene while Multipath [6] further set anchors to output multi-modal prediction and IntentNet [15] set a different motion predictor for different predefined intents. PRECOG [9] proposed a flow-based generative models conditioned on goals with CNN to process context. However, as pointed [4], those CNN-based methods' performances are significantly influenced by the receptive field, feature cropping strategy, and image resolution and it is very computationally expensive and inefficient compared to directly process the coordinate data. TPCN [7] conducted trajectory prediction from point-cloud perspective which is an interesting exploration across fields. Recently, GNN-based [3, 16, 17, 4, 18, 6, 19, 5] has gained great success and dominated most of the trajectory prediction datasets. Many of them [16, 6, 19, 5] set an ego agent and used it to normalize data. In the concurrently released Waymo Open Motion Dataset's paper [8], they implemented TNT [6] by the forward-N-times way to do multi-agent prediction. In ILVM[20], they proposed an end-to-end joint prediction framework, in which they first used CNN to extract features from LiDAR point clouds and raster map, used GNN to capture interactions, and used a novel one stage parallel sampling to do joint prediction. As for normalization, they used Rotated Region of Interest Align[21], which is specificlly designed for LiDAR point clouds/images to align. In the concurrent work Scene Transformer[22], they used Transformer for both spatial and temporal dimension, adopted mask strategies and done joint prediction. They normalized data simply by the ego agent.

## 3 Problem Formulation

Suppose there are $N$ agents in the same scene (they could be either vehicles or pedestrians), and our prediction task is to learn the joint distribution of their future trajectories conditioned on their historical states. Let us assume that the current time step is $t = 0$. For each agent $i = \{1, \cdots, N\}$, we have an observation of his/her states (such as coordinate, velocity, yaw angle, ...) from $t = -L$ to $t = -1$. We denote the joint observation as $\boldsymbol{X} = \{X_1, X_2, ..., X_N\}$ where $X_i = \{\boldsymbol{x}_i^{-L}, ..., \boldsymbol{x}_i^0\}$. Let us also define the future coordinates for all $N$ agents from $t = 1$ to $t = T$ as $\boldsymbol{Y} = \{Y_1, Y_2, ..., Y_N\}$ where $Y_i = \{\boldsymbol{y}_i^1, ..., \boldsymbol{y}_i^T\}$. Then, the joint prediction model is designed to learn the following conditional distribution:

$$p(Y_1, Y_2, ..., Y_N | X_1, X_2, ...X_N) \tag{1}$$

On the other hand, the single-prediction model is approximating a marginal distribution defined as follows:

$$p(Y_i | X_1, X_2, ...X_N) \quad \text{where agent i is the ego agent.} \tag{2}$$

In the cases of multiple-agent prediction, essentially, multiple-inference is using multiple marginal distributions to approximate the joint distribution by implicitly assuming the multiple marginal distributions are conditionally independent [8]:

$$p(Y_1, Y_2, ..., Y_N | X_1, X_2, ...X_N) = \prod_{i=1}^{N} p(Y_i | X_1, X_2, ...X_N) \tag{3}$$

---

[2]The generalization track is testing the model's performance on new maps/scenarios which have never been seen in the training time.

**Only Forward**

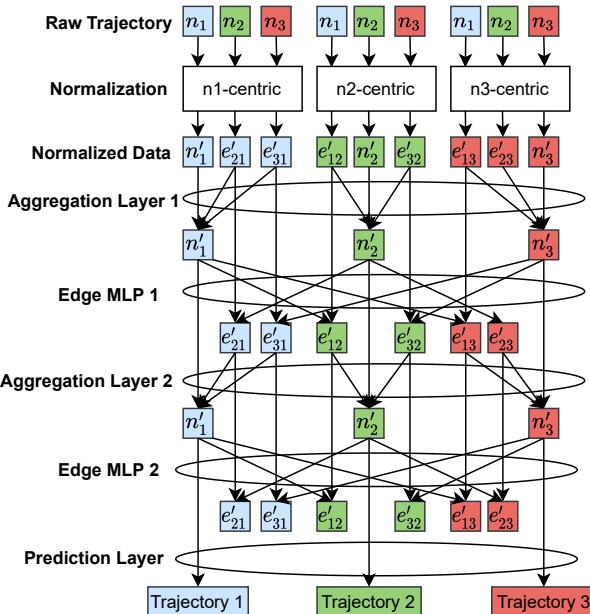

Figure 2: The structure of the proposed model: we use 3 agents as an example. $n_1, n_2, n_3$ means raw data. The superscript ′ means the data after normalization. $e_{ij}$ means the normalized data of agent $i$ by the origin and orientation of agent $j$. First, all agents are encoded in their egocentric views in the normalization step; second, the node features (Aggregation Layer) are updated by the egocentric information separately for different agents; and finally, the edge features (Edge MLP) are updated using allocentric views, summarizing local information from all agents.[3]

Unfortunately, such assumption is not true in practice, particularly when there are intensive interactions among the $N$ agents.

## 4 The Model

An overview of the model is shown in Fig. 2. The model has the following three key modules:

**View Transformation (Normalization Layer):** All agents are encoded symmetrically in the graph. In the normalization layer, we set each agent in his/her own egocentric view instead of selecting one centric agent and normalize all other agents based on it. Specifically, for the coordinate system of each node (agent) and its in-edges' (neighbors), we use the agents' coordinate at $t_0$ (w.r.o.g) as the origin and let the yaw angle be aligned with the negative x-axis. Since the input node and edge features are temporal sequences, we use a temporal sequence encoder (ResNet-18 1D version+MLP) to encode the sequence into a feature vector.

**Node update (Aggregation Layer):** Update each node feature according to their in-edges' features to incorporate neighbor's information $h'_i = f(\{h_{ji}, j \in \text{Neighbor(i)}\})$ where $h'_i$ is the updated node feature of agent i, $h_{ji}$ is the feature of in-edge from agent j to i, $f$ is the aggregate function and it is multi-head attention in Transformer. The Transformer is proposed in [23] which is a permutation-invariant operator and could capture the relations among a set of elements. Since all agents in the same scene are permutation-invariant, we directly adopts it. Due to the limited space and it is not our main contribution, for more details about Transformer, please refer to [23]. Note that for conciseness we added self-loop for each node which means each nodes' neighbors include itself. Since the input node feature and input in-edge features of each agent are both normalized in each agent' coordinate system, it is in the egocentric view.

---

[3]Note that the Edge MLP in the last layer could be removed to reduce computation (in this case - Edge MLP 2)

**Edge update (Edge MLP):** Update each edge by their source node's feature to obtain updated neighbors' information $h'_{ji} = g(h_{ji}, h'_j)$ where we denote the concatenation+FeedForwardNet in Transformer as $g$. Since the new information is coming from the source node - the neighbor in the neighbor's view, it is in the allocentric view.

We would like to emphasize the benefit of decoupling the node update step and the edge update step. By decoupling, the node serve as the **storage** for the interaction information between each agent and its neighbors in the egocentric view. Via such design, the edge feature can be efficiently updated to collect and summarize the allocentric information. Hence, such a combined egocentric-and-allocentric encoding and feature updating can help to 1) achieve an unbiased encoding of all agents for joint prediction, and 2) improve the efficiency of the computing process.

## 5 Experiements

### 5.1 Metrics

As for the metric, we adopt the widely used ADE and FDE. They are defined as: Average Displacement Error (ADE) represents the euclidean distance averaged over time and agents between the ground truth and the prediction. It is then averaged over all cases. The ADE of a single case is calculated as:

$$\text{ADE} = \frac{1}{NT} \sum_{n,t} \sqrt{(\hat{x}_{n,t} - x_{n,t})^2 + (\hat{y}_{n,t} - y_{n,t})^2} \tag{4}$$

where N is the number of target agents in the case, T ist the prediction horizon, $\hat{x}$ and $\hat{y}$ represent the groud truth coordinate, x and y represent the predicted coordinate. The unit is meter.

Final Displacement Error (FDE) represents the euclidean distance averaged over agents between the ground truth and the prediction at the last predicted time-step. It is then averaged over all cases. The FDE of a single case is calculated as:

$$\text{FDE} = \frac{1}{N} \sum_{n} \sqrt{(\hat{x}_{n,T} - x_{n,T})^2 + (\hat{y}_{n,T} - y_{n,T})^2} \tag{5}$$

where N is the number of target agents in the case, T ist the prediction horizon, $\hat{x}$ and $\hat{y}$ represent the groud truth coordinate, x and y represent the predicted coordinate. The unit is meter.

### 5.2 Results

We tested the performance of the proposed model against several SOTA models on two interaction-rich dataset: the INTERACTION dataset and the TrajNet++. Please refer to for experimental setting details.

In INTERACTION dataset [12], we compared with DESIRE [24], MultiPath [25], TNT [6], and ReCog [26]. The former three are compared in [6]. We also listed the results of the top-rank methods on the leader board of the INTERACTION dataset. As we can see Tab. 1, our method outperforms the baseline methods, especially on the generalization ability track. We achieved the $1^{st}$ **place** on both the regular test track and generalization ability track of the INTERACTION dataset.

In the TrajNet++ [13], existing results in this dataset are all measured only by the performance on the pre-defined 'interesting' primary pedestrian ignoring other agents. Thus, for fair comparisons, we re-implemented the two top-ranked methods under our multi-agent prediction setting: 1. SocialLSTM [1], a typical pooling-based method and the top-ranked method on TrajNet++'s leader board, is a good comparison for our GNN-based method. 2. D-LSTM [13], which was proposed in the TrajNet++ dataset paper and achieved SOTA performance in their paper, captures social interactions by concatenating top-4 nearest neighbors' hidden states. Note that [13] stated in their paper that predicting pedestrian trajectories other than the primary pedestrian's would dampen their results which aligns well with ourfollowing for the single-view model. On the other hand, the GNN-based methods should be further improved to tackle high-density data like CFF dataset, which can be handled by pooling-based or top-k-neighbor-concatenation-based method due to the low computational complexity.

Table 1: Comparison with SOTA methods on the INTERACTION Dataset. - means the results are not publicly available. * means the model is trained by smaller gap between snippets which means more data, which is proposed by the challenge winner ReCog [26]. . The unit of ADE and FDE are both meters. ↓ means the lower the better.

| | Val ADE/FDE↓ | Test ADE/FDEred↓ | Generalization ADE/FDE ↓ |
|---|---|---|---|
| DESIRE [24] | 0.32/0.88 | - | - |
| MultiPath [25] | 0.30/0.99 | - | - |
| TNT [6] | 0.21/0.67 | - | - |
| MIFNet | - | 0.1973/0.6641 | 0.5339/1.4248 ($2^{nd}$) |
| ReCog* [26] | 0.1919/0.6462 | 0.1878/0.6381 ($3^{rd}$) | 0.5539/1.9187 |
| Mix | - | 0.1826($1^{st}$)/0.6423 ($2^{nd}$) | - |
| Ours | 0.1723/0.5988 | 0.1903/0.6563 | 0.3394/1.1983 |
| Ours* | 0.1700/0.5927 | 0.1840($2^{nd}$)/0.6344 ($1^{st}$) | 0.3263/1.1426 ($1^{st}$) |

Table 2: Comparison with top-ranked methods on the TrajNet++ dataset under our multi-agent prediction setting. The unit of ADE and FDE are both meters. ↓ means the lower the better.

| | Val ADE/FDE ↓ |
|---|---|
| Social LSTM [1] | 0.2848/0.5604 |
| D-LSTM [13] | 0.2915/0.5738 |
| Ours | 0.2079/0.4270 |

# 6 Comparison with Variants of the Proposed Model

In this section, we compare the the proposed model against forwarding the single-prediction models N times (FNT) and its three variants in temrs of running speed and performance. By theoretical analysis and experimental results, we will show that the proposed model could achieve the best efficiency and accuracy among several settings of GNN for the multi-agent prediction.

## 6.1 Baseline - forwarding $N$ times (FNT)

For a fully connected graphs with $N$ agents, there are $\frac{N(N-1)}{2}$ edges, which means each GNN layer has $\frac{N(N-1)}{2}$ message passing and $N$ updating. Thus, the computational complexity of $L$ layers of GNN in one forward is $L * (O(\frac{N(N-1)}{2}) + O(N)) = O(N^2)$ since L is a hyperparameter and we could treat it as a constant. As a result, in Equ. 6, we derive the computational complexity of FNT with GNN-based model on the fully-connected graph.

$$\underbrace{O(N)}_{\text{Number of Forwards}} * \underbrace{O(N^2)}_{\text{Complexity of GNN on Fully Connected Graph}} = O(N^3) \tag{6}$$

In contrast, as shown in Fig. 2, for the proposed model, the computational complexity of the two consecutive steps (Aggregation Layer and Edge MLP) both are $O(N^2)$ which makes the overall complexity $O(N^2)$.

**Represent FNT by a single graph:** The incorporation of neighbors' information in GNN essentially is: each target node obtains source nodes' information through their in-edges. Thus, FNT is equivalent to: each agent transforms its own states and others' states into its ego-centric coordinate system by normalization. Its own transformed states could be seen as a node while others' transformed states could be seen as its in-edges. Then, for each agent, it maintains and updates both its node feature and in-edge features in GNN. Finally, it uses its node feature to do the trajectory prediction. Fig. 3a gives the illustration of above description.

Here, we conduct comparison experiments between FNT and the proposed methods. Experiments results are in Tab. 3. As we can see, compared to FNT, our method could significantly reduce the computational time while achieving better results.

Table 3: Comparison between the proposed method and forward-N-times method. The unit of ADE and FDE are both the meter. The unit of Train/Val time is the second. ↓ means the lower the better.

| | INTERACTION | | TrajNet++ | | |
|---|---|---|---|---|---|
| | Val ADE/FDE↓ | Train/Val Time ↓ | Val ADE/FDE ↓ | Train/Val Time ↓ | #Parameters |
| FNT | 0.1840/0.6231 | 1261s/50s | 0.2328/0.4659 | 762s/23s | 579991 |
| Ours | 0.1723/0.5988 | 713S/27s | 0.2079/0.4270 | 175s/10s | 678807 |

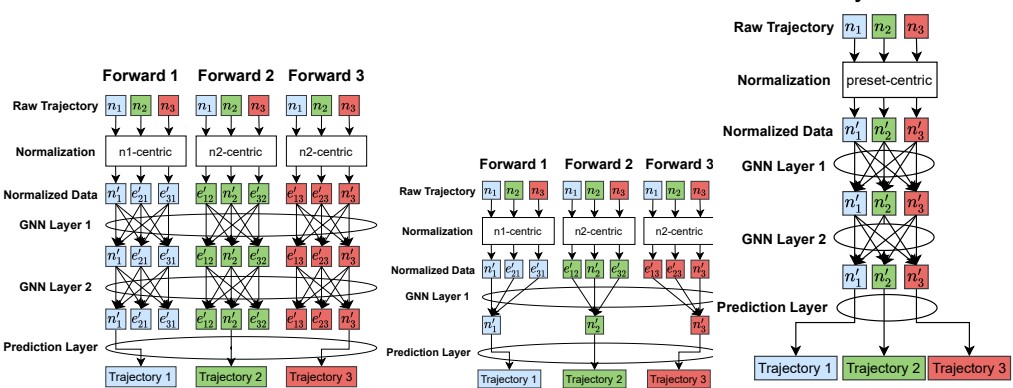

(a) Illustration of computational process of forward-N-times (FNT).

(b) Illustration of computational process of one-layer GNN (OL).

(c) Illustration of computational process of average of all agents (AA) and randomly selection (RA).

Figure 3:

## 6.2 Variant I - stop maintaining the edge feature

After finding out the abundance, we explore some other variants which could reduce the computational complexity of multi-agent prediction to $O(n^2)$ like the the proposed model and evaluate their precision.

Since the abundance is on the edges, the first intuitive variant is: **how about we stopping maintaining the edge feature and only update the node feature?** In this variant, there is no multi-hop or multi-order information included. For example, for agent 1, it could not obtain the interaction information between agent 2 and 3. Thus, this variant is equivalent to the one-layer GNN which only updates the node. We denote it as OL (one-layer). The illustration is in Fig. 3b. From the figure, we can obtain the computational complexity of OL in Equ. 7.

$$\underbrace{O(N)}_{\text{Number of Forwards}} * \underbrace{O(N)}_{\text{Ego Agnet Uses Its Neighbors' Features}} = O(N^2) \tag{7}$$

However, there might be two problems in one-layer GNN: 1: limited expressive power and hard to capture complex interactions. 2: for non-fully connected graph, lack of multi-hop neighbors' information.

**Validation Experiments** Here, we compare the results of GNN with one layer and two layers to evaluate the OL's influence on the models' performance. We compare the following settings:

- FNT: 2 layers GNN + fully-connected graph
- OL: 1 layer GNN + fully-connected graph
- FNT-Graph: 2 layers GNN + connected by distance threshold graph
- OL-Graph: 1 layer GNN + connected by distance threshold graph

For more experiment details, please refer to Appendix.

The results are in Tab. 4. We can draw the conclusions that 1. using fully-connected graph could obtain better results. Thus, we use fully-connected graph for the rest of experiments in the paper. 2. OL consistently performs worse than FNT.

In conclusion, **though not maintaining the edge feature could reduce the computational complexity, it loses much expressive power and high-order information which makes it impractical**.

## 6.3 Variant II - forwarding only once

Another option to reduce the computation is to only forward once, which means the model uses all nodes in the graph to do prediction instead of just using the ego node. As a result, the total computational complexity is reduced to the computational complexity of GNN - $O(N^2)$.

Table 4: Validation experiments for the one-layer GNN(OL) refinement. The unit of ADE and FDE are both the meter. The unit of Train/Val time is the second. ↓ means the lower the better.

| ADE/FDE | INTERACTION Val ↓ | TrajNet++ Val ↓ | #Parameters |
|---------|-------------------|-----------------|-------------|
| FNT | 0.1840/0.6231 | 0.2328/0.4659 | 579991 |
| OL | 0.1908/0.6379 | 0.2337/0.4753 | 481303 |
| FNT-Graph | 0.1909/0.6704 | 0.2334/0.4693 | 579991 |
| OL-Graph | 0.1991/0.6864 | 0.2343/0.4805 | 481303 |

Table 5: Validation experiments for the Average of Agents(AA) and Random Agent (RA) plan. The unit of ADE and FDE are both the meter. The unit of Train/Val time is the second. ↓ means the lower the better.

| | INTERACTION | | TrajNet++ | | |
|---|-------------|---|-----------|---|---|
| | Val ADE/FDE ↓ | Train/Val Time ↓ | Val ADE/FDE ↓ | Train/Val Time ↓ | #Parameters |
| FNT | 0.1840/0.6231 | 1261s/50s | 0.2328/0.4659 | 762s/23s | 579991 |
| RA | 0.1938/0.6474 | 1308s/51s | 0.2483/0.4833 | 776s/24s | 579991 |
| AA | 0.2187/0.7189 | 642s/21s | 0.2382/0.4762 | 141s/6s | 579991 |

However, another problem arises: **how to normalize data without setting an ego agent?** The following are two intuitive variants:

- Average of Agents (AA): we take the average coordinate and yaw angle of all agents at $t_0$ as the origin and the orientation of the positive x-axis respectively.

- Random Agent (RA): we randomly select an agent as the ego agent to do the normalization. Note that though it is faster in inference in theory, in training/validation, we still need to input all possible ego vehicle choices, which means no speed up. In fact, compared to FNT, it is equivalent to adding an auxiliary task of trajectory prediction for non-ego agents.

The illustration of AA/RA is in Fig. 3c. We compare the performance of RA and AA to the original forward-N-times (FNT) in Tab. 5. We also record the Train/Val time which is measured by the time to run one epoch on the train/val set on one RTX 2080 Ti with all GPU memory.

From the results, we can conclude that: 1. Forward only one time could significantly reduce the computational time. 2. However, the two alternative normalization methods dampen the performance. 3. AA performs the worst. We conjecture that AA is too sensitive since it is influenced by all agents in the scene and it cannot effectively reduce the input and output space. 4. TrajNet++ (pedestrian) dataset was less influenced. We conjecture that it is due to the smaller range of scenes for pedestrians.

In conclusion, **forward only once is effective for decreasing the computational complexity while a more stable normalization strategy is needed to maintain the performance.** Thus, as demonstrated before, in the proposed model, all agents to be treated symmetrically and homogeneously is significant for the performance of the model.

## 7 Conclusion

In this paper, we presented a multi-agent prediction model which combines the egocentric and allocentric views. It achieves SOTA performances on the two interaction-rich behavior datasets: INTERACTION (vehicles) and TrajNet++ (pedestrian). In the INTERACTION dataset's challenge, the model is ranked in the first place in both the regular track and generalization track. Besides, compared to the common multi-inference single-prediction model with similar architecture, the model could increase the running speed by 1.7-4.5 times.

In this paper, the Transformer and the Edge MLP only are chosen as a implementation to show the effectiveness of combining the egocentric and allocentric views. For future work, more specific designed structure could be explored. For example, to further reduce the computational complexity, some efficient variants of the Transformer could be explored. To better capture the relations of the two views, more complicated structure could be tried instead of the Edge MLP (maybe another Transformer). Additionally, we might extend it to incorporate the map information and do multi-modal prediction.

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
