# OpenReview forum: "Multi-Agent Trajectory Prediction by Combining Egocentric and Allocentric Views"
_robot-learning.org/CoRL/2021/Conference — CoRL2021 Poster_

### Official Review · Reviewer_CH6w · 2021-07-21

**Originality:** Good
**Technical Quality:** Good
**Clarity Of Presentation:** Very Good
**Impact:** 4

**Recommendation:**

Weak Accept: I recommend accepting the paper, but will not argue for my recommendation if the majority of other reviewers have a different opinion.

**Summary:**

The paper presents an efficient multi agent prediction framework that treats multiple agents as a unit, rather than as a composition of multiple single-agent predictions. Within the presented framework, the authors address problems with current methods for normalization and processing by considering all agents' information symmetrically considering both egocentric and allocentric perspectives.

**Issues:**

The main issue is that there is some mathematical rigor missing in the approach (model) section. This further inform why the model was constructed the way it was, and would improve the overall clarity and contribution of the work.

**Reviewer Expertise:**

Good: General knowledge of the area

**Strengths And Weaknesses:**

The paper was very clear and easy to understand. The problem was also well motivated in the problem formulation. The entire contribution was the network architecture, and I do believe that there could be more mathematical rigor used to explain the choices of the authors. The results were presented clearly- notably the efficiency analysis.

**Summary Of Recommendation:**

My main issue with the paper was that it needed more mathematical rigor in explaining some of the choices made about the specific model architecture, though I did appreciate the rigor in the section about efficiency. I think the organization is good and overall the idea of the paper is good. More explanation on why the model was unique would have assisted in the originality rating as well. The results overall show improvement and good results, so that's why it could be accepted.

---

> ### Author Response · Authors · 2021-08-31
> **Respose to Reviewer CH6w**
>
> We thank the reviewer for taking time to review the paper and giving helpful suggestions. We are glad to know that you like the written style, the concept about "egocentric" and "allocentric" and we appreciate your recognition! The following are responses to some comments and suggestions you made. We hope that they could solve your concerns. Please let us know if you want to know anything further or discuss about any questions and concerns further.
>
> 1. Comment: **My main issue with the paper was that it needed more mathematical rigor in explaining some of the choices made about the specific model architecture**
> - Response: We thank the reviewer for giving the concern. First, we would like to clarify that the core goal of this work is to find a multi-agent prediction framework with both high efficiency and good performance by exploring different normalization and message-passing strategies. We adopted widely-used structures to make sure that the comparison is based on the state-of-the-art techniques. With the structures being held as a constant, we compare the different normalization and message-passing strategies. By experiments, we confirmed that combining egocentric and allocentric views performs best. Second, we would like to clarify that using Transformer and MLP is an implementation of our insight. We think they are good choice because: (i) as for the information exchanging among variable number of elements, the Transformer has been proved to work well in many fileds. (ii) as for integrating information from different sources, concatenating the vectors and then feeding it into a MLP is one of the most commonly-used ways. It is simple yet effective. We agree that there might be some more specifically designed structures could perform better, but we believe that those commonly known structures are enough to demonstrate the effectiveness of the proposed model.
>
> 2. Comment: **More explanation on why the model was unique would have assisted in the originality rating as well.**
>  - Response: We thank the reviewer for the comments. We would like to clarify that to the best of our knowledge, our work is the first to consider the Egocentric and Allocentric view in the motion prediction problem. We believe this is an important insight. Also, we would like to emphasize that, currently, for the task of predicting multiple agents’ trajectories jointly, there is no standard normalizing strategy. For example, in the concurrent Waymo Motion Dataset paper (https://arxiv.org/abs/2104.10133), they achieve the joint prediction by running the TNT for each agent  (similar to FNT strategy in our paper). In the concurrent work - Scene Transformer (https://arxiv.org/abs/2106.08417), they set the ego agent as the reference (similar to RA strategy in our paper). Thus, in this paper, we would like to explore the best strategy to do normalization in the multi-agent setting. By experiments, we show that by incorporating the Egocentric and Allocentric views, the proposed method could achieve both fast speed and good performance.
>
> Thank you again for the reviewing and your recognition. We hope our response solves your concerns and questions.

---

### Official Review · Reviewer_Wu5R · 2021-07-23

**Originality:** Excellent
**Technical Quality:** Excellent
**Clarity Of Presentation:** Good
**Impact:** 4

**Recommendation:**

Strong Accept: I recommend accepting the paper and will argue for my recommendation even if other reviewers hold a different opinion.

**Summary:**

This paper considers the joint motion prediction of agents in a multi-agent system, motivated mostly by the prediction of pedestrians that could support safer autonomous driving vehicles. In contrast to fully decoupled methods, which propose to predict each moving agent (pedestrian) separately in a system, allowing great scalability at the cost of joint reasoning, this work proposes a novel combined ego- and allo-centric way of jointly predicting the motion of many pedestrians while keeping computation low. This work is demonstrated on a standard prediction benchmarking dataset against other SOTA methods, including other entries of an international competition, where it ranked first in two categories. Finally, many sub-variants of the work are presented and discussed, as a form of ablation study, to highlight the necessary features of the best method proposed.

**Issues:**

Issues were listed above in my main review (major and minor ones).

**Reviewer Expertise:**

Good: General knowledge of the area

**Strengths And Weaknesses:**

First of all, this paper reads rather well, with a clear goal, motivation, and premise. The paper is well structured, although I was a little taken aback by the prior work section being second to last: as a reader, I would have preferred to be given the prior work context before reading the rest of the paper so I could better situate this work. This is not a big deal, but I strongly recommend authors consider placing the prior work just after the introduction, as is very common in robotics/engineering papers.
One note that I have to make, is that there is an overwhelming number of small typos in this paper, which are very easy to miss but that I didn't, and which did impact my reading. I really recommend authors do a proper spellcheck of their work, as this is not OK for a published paper. I will provide a few below in my minor comments section, but I really believe this is easy to fix and should have been done before the initial submission.

Now, for more major issues, I do not have any concerns with the actual method described in this work, and I actually loved the idea of mixing egocentric and allocentric views to provide a symmetric, yet nuanced view of the world and improve predictions. The use of a GNN and transformers to allow scalability is smart, and was shown to lead to much better generalization (as evidenced by the ranking to that particular track of the challenge).
My only main issue would be with the presentation of most of the results in the paper: All tables have absolutely NO UNITS, making it impossible for the reader to understand anything about what they report. What are the numbers reported? Is higher/lower better? How are those results obtained, and what do they truly mean?
As a reader, I feel like I was told a beautiful story about a great new idea, but when it came to quantitative results I have no way of telling anybody anything about what performance I would expect from this method, and to why. This is not good, and should be expanded:
1. Authors need to explain how models are tested, and what metrics are reported (and higher/lower is better) before Table 1. This needs only be done once, as I guess all tables report values of the same type (again, this can only be a guess...).
2. There might be a few too many tables, and instead of one or two of them, I would have loved to see the actual results in some test cases. Can you show the actual ground truth of the pedestrians' motion, and then a comparison between your approach's predictions and that of other benchmarks? I loved the story about how decoupled prediction can affect prediction greatly, especially when pedestrians have strong inter-relations, and I would like to see that in some of the (visual) results.
3. I also worry that the second part of the paper is a little too lengthy for what it truly brings to the reader. There is a large discussion of complexity, and of those many variants of the proposed model, and I am left wondering if this really brought me a lot. Can this kind of "ablation study" be summarized in a more concise manner? In particular, do we really care about computational complexity for this work, i.e., is this really the main limitation that should drive the choice of variant of the model? I did see some differences in the tables, when the runtimes are discussed, but again due to the tables being near-impossibleto read I could not really appreciate why runtime/complexity was such a key point that would justify nearly 4 pages being dedicated to variants and complexity discussion.

Minor points:
1. Section 5.1, line 151: What does the "abundance" mean? This does not appear to be a standard term in the field (or any field I am familiar with or could google about), is this a typo/mistranslation or am I missing anything? If this is a standard term, please provide a citation or footnote to explain it, otherwise please correct this term.
2. Short list of some of the typos/grammatical mistakes I caught. Again, please perform a thorough grammatical/spell check, as the state of the text was really poor for a first submission:
	- line 30: "one for each agent" --> once for each agent
	- line 109: textbf1st place --> fix the missing backslash (latex typo)
	- line 119: modelOn --> missing period between end and start of two sentences.
	- Table 3 caption: Comparsion --> Comparison
	- line 142: Aggregateiton  --> Aggregation
	- Figure 4 caption: vairant --> variant
	- line 181-182: impracticable --> impractical?
	- line 200: AA performances the worst --> not an English sentence, this seems to be missing a verb

**Summary Of Recommendation:**

In summary, this work proposed a novel and intriguing new method to perform mostly-coupled, yet computationally tractable multi-agent motion prediction, by allowing agents to combine their view with that of other agents in the system. The paper reads well (minus some typos that can be easily fixed), and results seem strong (although presentation needs to be improved there as well). The use of the 8 pages is not optimal yet, with some sections needing to be made more concise so more visual results comparison can be added, but I believe this paper has a lot of potential. I am in favor of accepting this work for presentation at CoRL 2021, should my major and minor points be addressed.

---

> ### Author Response · Authors · 2021-08-31
> **Respose to Reviewer Wu5R**
>
> We thank the reviewer for taking time to review the paper and giving helpful suggestions. We are glad to know that you like the written style, the concept about "egocentric" and "allocentric" and we appreciate your recognition! The following are responses to some comments and suggestions you made. We hope that they could solve your concerns. Please let us know if you want to know anything further or discuss about any questions and concerns further.
>
> 1. Comment: **I was a little taken aback by the prior work section being second to last**
> - Response: We thank the reviewer for the suggestion about the paper writing and we have moved Related Works to Section 2.
>
> 2. Comment: **One note that I have to make, is that there is an overwhelming number of small typos in this paper, which are very easy to miss but that I didn't, and which did impact my reading.**
>  - Response: We thank the reviewer for the careful reading. We apologize for the inconvenience and we corrected typos in the updated manuscripts.
>
> 3. Comment: **Authors need to explain how models are tested, and what metrics are reported (and higher/lower is better) before Table 1.**
>  - Response: We thank the reviewer for the pointing out the unclear part. We are sorry for not giving the details about ADE/FDE in the main text due to the limited place. In the updated manuscripts, we added it to the main text Section 5.1 in the updated manuscripts. Additionally, we added many experimental and implementation details in the Appendix as well as some additional discussions and experiments. For your convenience, the unit of the metric ADE/FDE used all over the paper is the meter.
>
> 4. Comment: **Can you show the actual ground truth of the pedestrians' motion, and then a comparison between your approach's predictions and that of other benchmarks?**
>   - Response: We thank the reviewer for the suggestion. In Appendix A, we added visualization of the proposed model against the baseline methods under highly interactive scenarios of both INTERACTION and TrajNet++. From the visualizations and analysis, we could find that the proposed model could better capture the complex interactions among agents and make accurate predictions compared to baseline methods.
>
> 5. Comment: **I also worry that the second part of the paper is a little too lengthy for what it truly brings to the reader**
> - Response: We thank the reviewer for the suggestion. The reason why we used 4 pages to describe the variants is that as for predicting multiple agents’ trajectories jointly, there are no standard ways to do normalization. As in the Waymo Motion Dataset paper, they did it by Forward-N-Times(FNT). As in the concurrent work Scene Transformer (https://arxiv.org/abs/2106.08417), they did it only in the ego agent’s view (like RA model did). We would like to analyze and compare the popular and intuitive choices against the proposed model in terms of performances and speed. It could serve as a kind of ablation study to show whether and why decoupling node and edge updates in the proposed model could be better than those popular and intuitive ways. In fact, the experimental results show that the proposed method is the best strategy. In the updated manuscripts, we modified the title and descriptions of the section 6 to clarify the intention of the entire section.
>
> 6. Comment: **Section 5.1, line 151: What does the "abundance" mean?**
> - Response: We thank the reviewer for pointing out the unclear part. Here, the “abundance” means the unnecessary computing part in the Forward-N-Times (FNT) method. We have written a clearer description in the updated manuscript.
>
> Thank you again for the reviewing and your recognition. We hope our response solves your concerns and questions.

---

> > ### Comment · Reviewer_Wu5R · 2021-09-03
> > **Response to the Authors**
> >
> > Dear Authors,
> >
> > Thank you very much for your hard work updating the paper. Personally, I feel that my prior concerns have been adequately addressed.
> >
> > That being said, after reading the other reviews, and in particular that of reviewer ddzJ, I do believe that this new version of the work does need further writing work. In particular, I agree with reviewer ddzJ's points about some of the design choices not always being introduced very clearly and/or at the right place in the text, and sometimes needing some more justifications. Additionally, some of the claims (e.g., about computational complexity) likely need to be toned down to keep them perfectly accurate and avoid misleading the readers.
> >
> > However, provided these changes are made in the final camera-ready version of this work, I would like to keep my current recommendation for this paper.

---

> > > ### Author Response · Authors · 2021-09-04
> > > **Response to Reviewer Wu5R**
> > >
> > > Thank you for your response. We will update according to you and other reviwers' suggestions in the final camera-ready version.
> > >
> > > Thanks again for taking the time to review and giving advice to make the paper better!

---

### Official Review · Reviewer_zwfy · 2021-07-23

**Originality:** Good
**Technical Quality:** Very Good
**Clarity Of Presentation:** Excellent
**Impact:** 3

**Recommendation:**

Strong Accept: I recommend accepting the paper and will argue for my recommendation even if other reviewers hold a different opinion.

**Summary:**

This paper present a method for multi-agent prediction that is based on GNNs, but performs joint prediction by alternatively taking a local agent view as well as a view akin to "how do the other agents view each other, and me?" The paper is extremely well-written and intuitive, and this is improved by several visual aides, particularly figures 1 and 2. The method achieves good performance, both in terms of prediction and computational complexity. I encourage the authors to provide stronger justification for a few choices, as I try to detail in my comments below.



**Issues:**

I appreciate the early discussion about the terms "egocentric" and "allocentric", which will likely be quite helpful for the target audience at such a conference.

I appreciate the discussion in lines 36-41. However, I am not sure this is a limitation of GNNs; i.e. if the distinction between egocentric vs allocentric views is as critical as the authors suggest. For example, can't an agent identify proper scales between distances (and other spatial relations) for other agents given current sensing modalities? Perhaps the authors could tighten up these claims. In context of vehicles, it could be that they are all sharing state information via V2X technology, in which case obtaining an unbiased graph representation seems feasible, if not straightforward.

Intuitively, I like the ideas presented in lines 42-50, but (at this point in the reading) wonder about computational complexity and inference time.

I enjoy the qualitative thinking and "logic" presented in figure 1; this kindof thing can be quite valuable and I think it helps with readability.

To make the paper more clear and self-contained, the authors should define the important metrics of ADE and FDE. Furthermore, I assume the authors are using the same prediction horizon (12 future steps, given 9 past steps) as the baselines? I might have missed these details and apologize if I have, but this was not obvious from the writing.

Typo in line 142, "aggregateiton"

This is somehwat of a philophical question, but if some type of oracle has access to all information (i.e. agent 1 has access to n2 and n3 centric data), why not just create a weighted graph in the very beginning that represents a global, properly scaled set of relative spatial information (for example, edges are simply weighted by relatve distance, bearing, heading, etc)?

INTERACTION and TrajNet++ are perfectly fine datasets. However, there is one issue in the sense that (as the authors of TrajNet++ point out), not all of the relevant methods have even been evaluated on a consistent baseline. Those authors do well to be quite comprehensive, but I am not sure the results in your paper, e.g. in Table 2, represent a comprehensive comparison. There are many other methods besides Social LSTM (there is a typo in Table 2, by the way) and D-LSTM that have achieved good performance on data sets outside of TrajNet++, and furthermore, neither of these methods are explicitly based on GNNs. The suggestion here is to justify the choices of both the data used, as well as the baselines for comparison, so that this is all more rigorous and comprehensive. I recognize that this is nearly impossible (as the sparsity of table 1 indicates), but this could still be improved.

The authors seek to make a contribution to GNN-based methods of multi-agent prediction. As the authors acknowledge later, in section 6, not all methods are abased on on GNNs, or at least not explicitly. Social LSTM and GAN are such examples, as are several more recent works (I am trying to say that GNNs are not the only currently hot method of doing this). The authors would do well to justify the motivation for focusing on graph formulations, particularly in section 1 or 2, rather than having the reader wait (and try to infer this) based on section 6.





**Reviewer Expertise:**

Very good: Comprehensive knowledge of the area

**Strengths And Weaknesses:**

This paper was very well written - I sincerely appreciate the effort to bring intuition into the discourse, which unfortunately is not the general case in academic literature. It was actually a delight to read.

I have a few clarifying questions and comments in the "Issues" section below; otherwise, I do not have any substantive suggestions for improvement.

**Summary Of Recommendation:**

My main reasons for this recommendation have to do with relevance, readability, and soundness of the approach. The paper is relevant in two ways: multi-agent prediction is important for myriad robotic tasks, and the focus on GNNs is a very active topic. While I am not 100% convinced that this is an extremely original idea, I do think the notion of taking multiple views in the multi-agent setting is interesting.

---

> ### Author Response · Authors · 2021-08-31
> **Respose to Reviewer zwfy**
>
> We thank the reviewer for taking time to review the paper and giving helpful suggestions. We are glad to know that you like the written style, the concept about "egocentric" and "allocentric" and we appreciate your recognition! The following are responses to some comments and suggestions you made. We hope that they could solve your concerns. Please let us know if you want to know anything further or discuss about any questions and concerns further.
>
> 1.	Comment: **I appreciate the discussion in lines 36-41. However, I am not sure this is a limitation of GNNs; … if not straightforward.**
> -	Response: We thank the reviewer for the discussion about the unbiased graph representation. We agree that via V2X technology, accurate and synchronous state information could be shared among all vehicles. However, as for “can't an agent identify proper scales between distances (and other spatial relations) for other agents given current sensing modalities”, in this paper, the experimental results shows that if we try to let the neural network learn to identify it (like AA/RA variant did) instead of explicitly designing the neural network architecture by the prior knowledge about egocentric and allocentric views, the performance would drop.
>
> 2.	Comment:  **To make the paper more clear and self-contained, the authors should define the important metrics of ADE and FDE. Furthermore, I assume the authors are using the same prediction horizon (12 future steps, given 9 past steps) as the baselines**
> -	Response: We thank the reviewer for the pointing out the unclear part. We are sorry for not giving the definition of ADE/FDE in the main text due to the limited place. We added it to the main text Section 5.1 in the updated manuscripts. Additionally, we added visualizations, many experimental and implementation details in the Appendix as well as some additional discussions and experiments. For your convenience, we give the prediction horizon we used in the whole paper here:
> as for the prediction horizon in TrajNet++, the prediction horizon is 12 future steps given 9 past steps while in INTERACTION, the prediction horizon is 30 future steps given 10 past steps. They are both the official settings.
>
> 3.  Comment: **This is somehwat of a philophical question, but if some type of oracle has access to all information (i.e. agent 1 has access to n2 and n3 centric data) … a global, properly scaled set of relative spatial information (for example, edges are simply weighted by relatve distance, bearing, heading, etc)?**
> - Response: We thank the reviewer for sharing their idea. Actually, if we set a suggested graph (i.e. agent 1 has access to n2 and n3 centric data), then the number of edges would be O(N^4). The calculation is that there are N agents in each centric data. There are N different centric data. If all the $N\times N$ agents in all centric data are connected to each other, there are (N*N)^2 = N^4 edges, which is too complicated. In the paper, we have shown that even O(N^3) edges are unnecessary.
>
> 4. Comment: **However, there is one issue in the sense that (as the authors of TrajNet++ point out), not all of the relevant methods have even been evaluated on a consistent baseline … The suggestion here is to justify the choices of both the data used, as well as the baselines for comparison, so that this is all more rigorous and comprehensive**
> - Response: We thank the reviewer for the suggestion and understanding the difficulty of implementing the baseline methods.
> -- As for INTERACTION dataset, the only difference between ours and the official split is that we predict all agents in a snippet at once while the official split treats each agent as a single case. Since the ADE/FDE metric is averaged by agents, we could directly compare to other baselines.
> -- As for TrajNet++, since our multi-agent prediction setting measures the average of all agents’ ADE/FDE while the official TrajNet++ only measures the ADE/FDE of the primary agent, the results reported in other papers cannot be used directly. Thus, we re-implemented two top-ranked models on their leaderboard. As for the comparison with GNN based models, the FNT could serve as a representative for those most common message-passing GNN models.
> Thanks again for your suggestion. We added those clarifications in the Appendix D.1.
>
> 5. Comment: **The authors would do well to justify the motivation for focusing on graph formulations, particularly in section 1 or 2, rather than having the reader wait (and try to infer this) based on section 6.**
> - Response: We thank the reviewer for the suggestion about the paper writing and we have moved Related Works to Section 2.
>
> Also, we thank the reviewer for the careful reading and we corrected typos in the updated manuscripts.
>
> Thank you again for the reviewing and your recognition. We hope our response solves your concerns and questions.

---

### Official Review · Reviewer_ddzJ · 2021-07-24

**Originality:** Fair
**Technical Quality:** Fair
**Clarity Of Presentation:** Fair
**Impact:** 2

**Recommendation:**

Weak Reject: I recommend rejecting the paper, but will not argue for my recommendation if the majority of other reviewers have a different opinion.

**Summary:**

The paper presents an approach to predict trajectories of multiple agents. The paper attempts to motivate its model as capturing of "Egocentric View" and "Allocentric View", although the technical details on how interactions between such things could be captured is missing. A particular architecture is proposed without comparison to baselines or alternatives. The proposed model is a combination of Transformers and MLPs (on top of 1-D ResNet stems with different inputs).

**Issues:**

The authors are encouraged to use robot data (or at least robot sim) and better provide technical details and justification behind the proposed approach. As mentioned above, baseline experiments and ablations to justify the claim will help greatly.


**===================Updated review and suggestion after the discussion phase===================**

I really would like to give the accept rating to this paper as it seems to have a certain contribution, but the concerns I have overweigh it. The discussion with the authors partially addressed some of the initial concerns but it simultaneously also added others.

The authors clarified that the paper is introducing the layer normalizing representations based on multiple views. The interaction with the authors also clarified that the rest of the approach is a Transformer application (and MLP) on top of them. This is one design choice adopting layers, and it is OK as it showcases the benefits of the normalization layer. The authors also have done some experimental comparison to the baselines and ablations, although it could have been against stronger approaches.

The concerns I have are as follows, which I believe should be addressed in order for the paper to be ready in a publishable form.

- The paper should be more upfront about the fact that its "aggregation layer" exactly is a Transformer layer over the outputs of their normalization layer. The Transformer paper [26] is being cited only in Section 6, although the paper starts using Transformer much earlier in the paper. [26] is not even being introduced or discussed in the related work section, which will confuse and mislead the readers. The paper should introduce the Transformer paper properly early in the paper so that the readers can understand what it is, and the paper should explicitly point out how it relates to their approach.

- The interaction with the authors revealed the stance that they want to treat any architecture different from theirs as an "extension" of theirs, which we believe is an overclaim and academically misleading. The paper is introducing one particular design (which is perfectly fine as mentioned above), not the best or optimal design. This has to be clarified in the final version.

- Computational complexity advantage claims in the ablations need to be toned down, as the experimental comparison is only done with weaker baselines designed by the authors. There could be better alternatives with other efficient Transformers, which the paper has not explored.

- The new table specifying the # of parameters of the models need to be added in the final version of the paper. This would allow the readers to judge themselves how many more parameters the proposed approach is using compared to the baselines to get a higher accuracy.

Overall, the paper should be more upfront about what exact technical component it is adopting from the previous papers and where their computation advantage is coming from (i.e. Transformers).

**Reviewer Expertise:**

Very good: Comprehensive knowledge of the area

**Strengths And Weaknesses:**

Strengths

- Unclear

Weaknesses

- The 'model' presented in the paper lacks technical details and it is difficult to understand how the proposed model captures the properties regarding "Egocentric View" and "Allocentric View" mentioned in the introduction. We are not convinced that applying a series of these 'Aggregation Layers' and 'Edge MLP' are sufficient to capture what is claimed in the paper.

- The technical contribution of the paper seems to be limited to a designed model combining existing Transformer layers and MLP. The technical originality is limited and incremental.

- I think the key thing missing is the baseline experiments, ablation, and visualizations to convince that the proposed method captures egocentric and allocentric information well, and that it does it better than naive baselines such as simply concatenating all such egocentric and allocentric features in the beginning. The paper proposes a particular architecture, and we are not sure whether this is the best way to enable what's claimed compared to alternative baselines.

- It is unclear how this paper is related to robot learning. The paper mentions that trajectory prediction in general is needed for autonomous driving, but it does not provide any details how it could realistically be used for the vehicles. INTERACTION Dataset used in the paper is composed of vehicle trajectories from a top-down view and TrajNet++ is based on pedestrian trajectories obtained from a top-down view, which are from very different views compared to the first-person view of an autonomous vehicle.

**Summary Of Recommendation:**

The mismatch between what the introduction is arguing and the model details makes it very difficult to understand what the paper is trying to achieve. What the paper needs to show is that the proposed method really captures properties claimed in the introduction experimentally, by comparing against different baselines and visualizing it. Further, we question whether its lacking connection to robot learning data really makes the paper sufficiently suitable for CoRL.

---

> ### Author Response · Authors · 2021-08-31
> **Respose to Reviewer ddzJ**
>
> We thank the reviewer for taking time to review the paper and giving helpful suggestions. The following are responses to some comments. We hope that they could solve your concerns. Please let us know if you want to know anything further or discuss about any questions and concerns further.
>
> 1.	Comment: **The 'model' presented in the paper lacks technical details and it is difficult to understand how the proposed model captures the properties regarding "Egocentric View" and "Allocentric View" mentioned in the introduction.**
> -	Response: We thank the reviewer for the suggestions regarding the technical details. Due to the limited space, we could not add to the main text. In the updated manuscripts, we added experimental and implementation details in the Appendix D. Especially in Fig. 5, Appendix D, we gave all the computational functions of the proposed framework.
>
> 2. Comment: **We are not convinced that applying a series of these 'Aggregation Layers' and 'Edge MLP' are sufficient to capture what is claimed in the paper.**
> - Response: We thank the reviewer for the suggestions about better illustrating the effects of the proposed model. In Appendix A, we added visualization of the proposed model against the baseline methods under highly interactive scenarios. From the visualizations and analysis, we could find that the proposed model could better capture the complex interactions among agents and make accurate predictions compared to baseline methods.
>
> 3. Comment: **The technical contribution of the paper seems to be limited to a designed model combining existing Transformer layers and MLP.**
> - Response: We thank the reviewer for giving your concerns. We would like to clarify that using the Transformer is an implementation of our insight in the GNN part. While we do not claim transformers to be our inventions, we think using the Transformer is a most proper model design for our problem.
>
> 4. Comment: **The technical originality is limited and incremental.**
> - Response: To the best of our knowledge, our work is the first to consider the Egocentric and Allocentric view in the motion prediction problem. We believe this is an important insight. Also, the experimental results on two popular datasets also show that the proposed model could achieve state-of-the-art performance.
>
> 5. Comment: **think the key thing missing is the baseline experiments, ablation, and visualizations**
> - Response: We thank the reviewer for the suggestions. We would like to clarify for each aspect:
> -- Baselines: We have compared the proposed model with top-rank models of two interaction-rich behavior datasets. Our model could achieve state-of-the-art performance.
> -- Ablation Studies: in section 6, we have implemented several variants. The results showed that the proposed method achieved best speed and performance among all variants, which verified the importance of our design.
> -- Visualizations: We added visualizations of the proposed model in the Appendix A to show the effectiveness of the proposed model.
>
> 6. Comment: **does it better than naive baselines such as simply concatenating all such egocentric and allocentric features in the beginning**
> - Response: We would like to clarify that the terms “egocentric” and “allocentric” represent different views of the information and in the proposed model they are implemented by different modules. There are no such things called egocentric and allocentric features and concatenating is not feasible as well.
>
> 7. Comment: **It is unclear how this paper is related to robot learning.**
> - Response: We thank the reviewer for giving the questions. We would like to clarify that autonomous driving is a sub-area of robotics. Since our paper is learning-based trajectory prediction, it is robot learning. Actually, there are plenty of autonomous driving related papers accepted by CoRL 2020.
>
> 8. Comment: **The paper mentions that trajectory prediction in general is needed for autonomous driving, but it does not provide any details how it could realistically be used for the vehicles. INTERACTION Dataset used in the paper is composed of vehicle trajectories from a top-down view and TrajNet++ is based on pedestrian trajectories obtained from a top-down view, which are from very different views compared to the first-person view of an autonomous vehicle.**
> - Response: We thank the reviewer for giving the questions. We would like to clarify that in a self-driving system, perception and prediction are usually decoupled. Our method can be directly incorporated into the prediction module. The prediction results could be used by the downstream modules like the planning module. It is also a common practice for self-driving system to make predictions in the top-down view. Actually, most of the works cited in the Related Works section are in the top-down view.
>
> Thank you again for the reviewing. We hope our response solves your concerns and questions.

---

> > ### Comment · Reviewer_ddzJ · 2021-08-31
> > **architecture ablations and comparisons**
> >
> > What we are trying to ask is a fundamental question: is the proposed architecture really necessary, or is the egocentric normalization good enough for your objective as long as we have a sufficiently large model.
> >
> > In this context, what was meant with a naive baseline concatenating "egocentric and allocentric features" is a simple architecture of merging results of 'n-centric' normalizations in the beginning of the network. If a concatenation is not possible due to the number of agents not being fixed, we can use Transformer layers to merge them similar to the proposed model.
> >
> > The authors tried to test something similar with the models in Fig 3 (b)(c) and Table 4, but it simultaneously is not fair as the models in Fig 3(b)(c) seems to use less number of layers with fewer parameters than the proposed model (please clarify with the exact FLOPS). Smaller models obviously will perform worse. In order to make a fair comparison, it is better to increase the number of layers/parameters in the baseline models to match that of the proposed architecture.
> >
> > It will be great if the authors can clarify.
> >
> > Another relevant concern in terms of the computation speed is that this could be highly dependent on the hardware. Could we authors please also report FLOPS when comparing models?

---

> > > ### Author Response · Authors · 2021-08-31
> > > **Response to architecture ablations and comparisons**
> > >
> > > We thank the reviewer for the quick response and detailed comments. The following are our responses for your new questions:
> > >
> > > 1. Comment: **what was meant with a naive baseline concatenating "egocentric and allocentric features" is a simple architecture ... similar to the proposed model**
> > > - Response: We thank the reviewer for proposing their idea about the implementation. First, as stated in your comments, the idea would be similar to the proposed model. Second, please allow us to demonstrate the computational complexity of the idea: there are N agents in each centric data. There are N different centric data. If all the agents in all centric data are connected to each other, there are (N*N)^2 = N^4 edges. In the paper, we have shown that even O(N^3) edges (FNT) are unnecessary. Thus, it is not a simple naive baseline but a much more complicated variant of the proposed model. In fact, the O(N^4) setting is not feasible for the 2080Ti even with batch size 1 for the dataset we used. Please understand us that it is difficult to do experiments with extremely huge models under our current equipment. We understand this might be a powerful model considering its huge computational complexity, but in this paper we would like to present a model with low computational complexity while maintaining the performance.
> > >
> > > 2. Comment: **The authors tried to test something similar with the models in Fig 3 (b)(c) and Table 4, ... it is better to increase the number of layers/parameters in the baseline models to match that of the proposed architecture**
> > >   - Response: We thank the reviewer for giving their concerns.
> > > As for Fig 3(c) - the RA/AA variants, we would like to clarify that what is shown in the picture is the computational graph. Since the parameters of GNN are shared for message passing and aggregate function, RA/AA has exactly the same number of parameters with Fig 3(a)- FNT.
> > > As for Fig 3(b) – the OL (one layer GNN), it indeed has less parameters. First, we would like to clarify that as stated Appendix D3, we train each model for 300 epochs and report the best results on the val set. We would like to add that at the end of training, all models have overfit a lot (train loss decreases while val loss increases). Thus, all models have enough capacity for the training data. Second, we understand that you want to compare with similar number of parameters. In fact, we have conducted the experiments before. The proposed model and FNT are both two-layer GNN with hidden dimension 128. We give the results for the one-layer (OL) GNN with hidden dimension 256 below.
> > > | ADE/FDE | INTERACTION Val | TrajNet++ Val|
> > > | ----------- | ----------- | ----------- |
> > > | OL (H=128)| 0.1908/0.6379 | 0.2337/0.4753 |
> > > | OL (H=256)| 0.1913/0.6367 | 0.2367/0.4810 |
> > > | FNT | 0.1840/0.6231 | 0.2328/0.4659 |
> > > | Ours | 0.1723/0.5988 | 0.2079/0.4270 |
> > > As we can see, it has no significant difference and we conjecture that all models with hidden dimension 128 already have more than enough capacity and thus additional parameters would not make much difference. Third, we would like to also point out that as stated in Line 223 of the updated manuscript, one-layer (OL) loses much expressive power and high-order information. Its poor performance comes from the fact that its design could not fully utilize the power of deep learning – stacking depth.
> > >
> > > Regarding the comparison of number of computations, please check next comment.
> > >
> > > 3. Comment: **Another relevant concern in terms of the computation speed is that this could be highly dependent on the hardware. Could we authors please also report FLOPS when comparing models**
> > >   - Response: We thank the reviewer for giving their concerns. Note that GNN aims to handle non-Euclidian data unlike CV (fixed size images by resizing) or NLP (fixed length sentences by padding). FLOPs is not a meaningful metric since the input graph could have arbitrary size. Thus, it is a common practice to give the running time of GNN for one epoch on a specific dataset since different datasets have different statistics of graph size. Please check the official benchmark page of DGL https://docs.dgl.ai/performance.html and you can find that they also used the running time on different datasets to compare the speed.
> > >
> > > However, we totally understand the reviewer’s concern about the fair comparison of speed. We have tried our best to control other possible variables. As stated in the Line 237-238, we record the Train/Val time which is measured by the time to run one epoch on the train/val set on one RTX 2080 Ti with all GPU memory. More specifically, we used the same server, the same GPU, and the same backbone to test the time. The only difference of code is the GNN function part which is the part we want to compare. Also, our theoratical analysis and experimental result align well when comparing O(N^2) with O(N^3).
> > >
> > > We hope our response solves your concerns and questions.  Please let us know if there are any other questions and concerns.

---

> > > > ### Comment · Reviewer_ddzJ · 2021-08-31
> > > > **Follow up questions**
> > > >
> > > > We thank the author for clarifying, but there are follow up concerns.
> > > >
> > > > 1. "Naive" baseline: We do understand that there are N^2 number of centric features, but consideration of all N^4 pairs are unnecessary depending on which layer we are using to merge. One might naively have a layer that merges N features per view (N^2), and then merge them across views (another N^2). There are also (computationally) more efficient Transformer mechanisms such as Performers that avoid quadratic attention computation, which will allow us to consider all N^2 number of features directly.
> > > >
> > > > 2. Could the authors simply report the total number of parameters in each model? It is confusing in its current form. We believe the total # of parameters should be a constant unlike FLOPS?
> > > >
> > > > There are lots of observations and discussions in ML that over-parameterization is helpful both computational and statistically, and what we are suggesting is to confirm that the advantage of the proposed model does not come from simple over-parameterization

---

> > > > > ### Author Response · Authors · 2021-08-31
> > > > > **Response to Follow up questions**
> > > > >
> > > > > We thank the reviewer for the careful reviewing and thinking. The following are our responses for your new questions:
> > > > >
> > > > > 1. Comment: **Naive baseline: We do understand that there are N^2 number of centric features, but consideration of all N^4 pairs are unnecessary depending on which layer we are using to merge. One might naively have a layer that merges N features per view (N^2), and then merge them across views (another N^2). There are also (computationally) more efficient Transformer mechanisms such as Performers that avoid quadratic attention computation, which will allow us to consider all N^2 number of features directly**
> > > > > - Response: We thank the reviewer for updating their idea. We would like to point out the overall computational complexity of the updated idea is still O(N^3). As for step 1 merging feature per view, you have N views and within each view you have N features. To update N features of each view, you need them to exchange information (within the same view) between each other so that they can be updated – O(N^2). Thus, the overall computational complexity of step 1 is O(N)*O(N^2). Similarly, as for step 2 – merging features across views, you have N feature and for each feature you have N views. To update N views of each feature, you need them to exchange information (for the same feature) between each other so that they can be updated – O(N^2). Thus, the overall computational complexity of step 2 is O(N)*O(N^2) as well. In fact, this updated idea could be seen as FNT with an additional layer to exchange information across views – i.e., double depth within each GNN layer. The reason why it is still a O(N^3) method is that you still need updating all the features in each step. Though your updated idea makes it O(N^3) instead of O(N^4), it is still not a O(N^2) solution.
> > > > >
> > > > > At this point, we think we may both agree that your idea is not a naive baseline at all. Instead, it is a potential extension of the proposed method, which we believe is not necessary to illustrate in this paper. In this work, we would like to verify the idea of egocentric and allocentric views and give an efficient and intuitive implementation of this insight. So, if you want to explore more advanced implementations about egocentric and allocentric views, we are glad to offer help. Also, we think your idea about incorporating more efficient Transformer is also great. Please feel free to contact us if you want to discuss further about your idea based on the egocentric and allocentric views.
> > > > >
> > > > > 2. Comment: **Could the authors simply report the total number of parameters in each model? It is confusing in its current form. We believe the total # of parameters should be a constant unlike FLOPS**
> > > > > - Response: For your new request, we give the table about the number of parameters below:
> > > > > | Model | # Parameters |
> > > > > | ----------- | ----------- |
> > > > > | OL (H=128)| 481303 |
> > > > > | OL (H=256)| 1904695 |
> > > > > | RA/AA/FNT | 579991 |
> > > > > | Ours | 678807 |
> > > > > As we can see, there is no significant difference between RA/AA/FNT and the proposed model. As for OL, we have shown on the table below that with more parameters it performs even slightly worse. We totally understand your concern about the over-parameterization and we have made sure this is not the case. Actually, in the very early exploration of this work, we have conducted experiments to find the most suitable hyper-parameters for the experiments so that we could make them a constant instead of a variable in the further experiments and thus they would not influence the comparison between different GNN settings.  We have tried much larger hidden dimension (256/512) and deeper GNN (3/4/5/6) for FNT. We do not see any significant gains or drop, which means the capacity of current setting is far more than enough. Thus, we choose the most efficient settings to conduct all sets of experiments in the paper (hidden-dim 128 and GNN-layer 2 if not specified). Additionally, for your information, in the coordinate-based trajectory prediction method with GNN, it is adopted by many methods to use this level of hidden-dim and GNN-layer (for example, the ReCoG/TNT baseline methods in this paper).
> > > > >
> > > > > We hope our response solves your concerns and questions. Please let us know if there are any other questions and concerns.

---

> > > > > > ### Comment · Reviewer_ddzJ · 2021-08-31
> > > > > > **Is the contribution and the scope of the paper in introducing the "Normalization Layer" and presenting one particular model to take advantage of the results of the normalization layer?**
> > > > > >
> > > > > > 1. Naive Baseline: We agree that this O(N^3) baseline is a different approach, but we disagree that this is a potential extension of the proposed method. Rather, this is an alternative as it is not using any of aggregation layer or edge MLP that comprises 2/3 of the proposed model. It will only share the normalization layer.
> > > > > >
> > > > > > If the contribution and the scope of the paper is in introducing its view "Normalization Layer" and presenting one particular model to take advantage of the outputs of the normalization layer, we agree to this. If it is the case, the paper should clarify so, as it is difficult to disentangle its three different components in the model in its current form.
> > > > > >
> > > > > > We understand that the authors have no intention to try additional baselines or more efficient attention models themselves due to the time constraint.
> > > > > >
> > > > > > 2. We believe it is up to the readers to decide whether the use of, for example, 85% parameters is not significantly different or not, and we suggest the authors to include this information in the final version of the paper.

---

> > > > > > > ### Author Response · Authors · 2021-09-01
> > > > > > > **Response to Is the contribution and the scope of the paper in introducing the "Normalization Layer" and presenting one particular model to take advantage of the results of the normalization layer?**
> > > > > > >
> > > > > > > We thank the reviewer for the suggestions. The following are our responses for your new comments:
> > > > > > >
> > > > > > > 1. Comment: **Naive Baseline: We agree that this O(N^3) baseline is a different approach, but we disagree that this is a potential extension of the proposed method. Rather, this is an alternative as it is not using any of aggregation layer or edge MLP that comprises 2/3 of the proposed model. It will only share the normalization layer. If the contribution and the scope of the paper is in introducing its view "Normalization Layer" and presenting one particular model to take advantage of the outputs of the normalization layer, we agree to this. If it is the case, the paper should clarify so, as it is difficult to disentangle its three different components in the model in its current form**
> > > > > > > - Response: We are glad that we both agree that it is not a naïve baseline. However, we would like to point that “this is an alternative as it is not using any of aggregation layer or edge MLP that comprises 2/3 of the proposed model.” is wrong. The step 1 of your proposed idea is exactly the Aggregation Layer we proposed in the paper. You proposed to use step 2 to aggregate features across views while we used Edge MLP to do so. Our model decoupled the updating of the nodes’ and edges’ features with the Aggregation Layer and Edge MLP of different views. That’s why it could achieve speed boost O(N^2) compared to the O(N^3) method while achieving even better performance compared to the O(N^3) method - FNT. Thus, we argue that the combination of all parts of the proposed model is one of our contributions instead of “one particular model to take advantage of the outputs of the normalization layer” as you suggested. Also, we thank the reviewer for understanding that implementing a more computational complicated model and trying efficient Transformer variants are not within the intention of this paper, which we did not claim as our contribution.
> > > > > > >
> > > > > > >
> > > > > > > 2. Comment: **We believe it is up to the readers to decide whether the use of, for example, 85% parameters is not significantly different or not, and we suggest the authors to include this information in the final version of the paper**
> > > > > > > - Response: We thank the reviewer for the suggestion. We will add it to the final version of the paper. Also, we would like to point out that the baseline OL (H=256) has 280% parameters but still worse performance, which shows that those models have more than enough capacity. We hope the reviewer understand that it is nearly impossible to make the number of parameters totally equivalent. In the experiments, we made sure all the hyper-parameters are the same and the number of parameters are at the same magnitude, which is a common experiments setting in the deep learning field (for example, SENet compared to ResNet).
> > > > > > >
> > > > > > > We hope our response solves your concerns and questions. Please let us know if there are any other questions and concerns.

---

> > > > > > > > ### Comment · Reviewer_ddzJ · 2021-09-01
> > > > > > > > **applications of standard Transformers**
> > > > > > > >
> > > > > > > > What I am arguing here is that there are multiple different possible architectures to capture information after the normalization layer, and the proposed method is just one of it. The use of standard Transformers to combine the outputs of the normalization certainly is a contribution of the paper, but it is a Transformer application and there are many good/bad alternatives to be explored further (unless the authors found the proposed architecture through optimization process such as architecture search). If the authors disagree to this general statement, it will be great if they can please do so explicitly.
> > > > > > > >
> > > > > > > > What the authors call "aggregation layer" is a very standard Transformer over all possible inputs. And what was suggested as the "naive baseline" is also a use of the standard Transformer layer, with inputs per view and then inputs across views. If the author's argument is that any Transformer layer over the representations across views should be called an "aggregation layer", please do so explicitly, as this will lead to another discussion.

---

> > > > > > > > > ### Author Response · Authors · 2021-09-01
> > > > > > > > > **Response to applications of standard Transformers**
> > > > > > > > >
> > > > > > > > > We thank the reviewer for the quick response. the following are our responses for your new comments:
> > > > > > > > >
> > > > > > > > > 1. Comment: **What I am arguing here is that there are multiple different possible architectures to capture information after the normalization layer, and the proposed method is just one of it. The use of standard Transformers to combine the outputs of the normalization certainly is a contribution of the paper, but it is a Transformer application and there are many good/bad alternatives to be explored further**
> > > > > > > > > - Response: As for the variants of the Transformer, as we stated before, we agree with your point that there might be some other good/bad alternatives. In this work, we choose the standard Transformers as an implementation of our insight about egocentric and allocentric views. We thank the reviewer for proposing such an interesting idea and we would like to explore these possibilities in the future work.
> > > > > > > > >
> > > > > > > > > 2. Comment: **If the author's argument is that any Transformer layer over the representations across views should be called an "aggregation layer", please do so explicitly, as this will lead to another discussion**
> > > > > > > > > - Response: We think there might be some misunderstandings. What we want to argue is that: it is unfair to state that “this is an alternative as it is not using any of aggregation layer or edge MLP that comprises 2/3 of the proposed model” and “If the contribution and the scope of the paper is in introducing its view "Normalization Layer" and presenting one particular model to take advantage of the outputs of the normalization layer”. Your idea’s first step - merging feature per view is the aggregation layer, which means this “1/3” is proposed in the paper. As for your idea’s second step – merging feature across views, we did not claim it is  “an aggregation layer". Here, the proposed model used Edge MLP – O(N^2) and your idea used the Transformer O(N^3), which is why we called your idea a potential extension. We understand your point that if using some kind of efficient Transformer, your idea could be O(N^2) and we would like to explore that in the future work.
> > > > > > > > >
> > > > > > > > > We hope we would reach an agreement that your idea is interesting and worth explorations in the future work while in this paper we focus on presenting the idea about the efficient structure to combine egocentric and allocentric views. Please let us know if there are any other questions and concerns.

---

> ### Author Response · Authors · 2021-09-12
> **Reply to updated review and suggestion after the discussion phase**
>
> We thank the reviewer for the updated suggestions.  The following are our plans to update the paper according to your kind suggestions.
>
> 1. Suggestion: **The paper should be more upfront about the fact that its "aggregation layer" exactly is a Transformer layer over the outputs of their normalization layer. … and the paper should explicitly point out how it relates to their approach**
> - Updating Plan: We will clarify in the updated paper by introducing Transformer earlier with more details and how it relates to the proposed method.
>
> 2. Suggestion: **The paper is introducing one particular design (which is perfectly fine as mentioned above), not the best or optimal design. This has to be clarified in the final version**
> - Updating Plan: We will clarify in the updated paper that we adopt the Transformer in Aggregation Layer and MLP in the Edge Updating Layer as an implementation of the proposed GNN which combines egocentric and allocentric views. We will clarify that the exploration of other design is a potential direction to improve the proposed method for the future work.
>
> 3. Suggestion: **Computational complexity advantage claims in the ablations need to be toned down, as the experimental comparison is only done with weaker baselines designed by the authors. There could be better alternatives with other efficient Transformers, which the paper has not explored**
> - Updating Plan: We will clarify that the exploration of the efficient Transformer which might significantly boost the speed of the GNN is a potential direction to improve the proposed method for the future work.
>
> 4. Suggestion: **The new table specifying the # of parameters of the models need to be added in the final version of the paper. This would allow the readers to judge themselves how many more parameters the proposed approach is using compared to the baselines to get a higher accuracy**
> - Updating Plan: We will add the # Parameters to the table in the final version of paper.
>
> Thank you again for providing your helpful suggestions to make the paper clearer. We totally understand your concerns and we will clarify those parts in the final version of paper. We hope our updating plan could solve your concerns. If there are any other suggestions, please let us know.

---

### Meta-Review · Area_Chair_4aGL · 2021-08-13

**Recommendation:** Accept (Poster)
**Confidence:** 4

**Metareview:**

This paper proposes a novel method for multi-agent prediction. While the reviewers are generally positive about this paper, they also raise a number of important concerns (in particular reviewer ddzJ). The authors should carefully address the reviewers' comments in their rebuttal. Also, I would encourage the authors to compare their work with this paper: https://arxiv.org/pdf/2007.12036.pdf

UPDATE POST DISCUSSION PHASE: I would like to thank the authors for their comments and clarifications during the discussion phase. This paper was extensively discussed by the reviewers and the Area Chair. In general, there is agreement that this paper provides a useful contribution. Yet, concerns persist regarding a number of aspects of this paper, in particular regarding the rigor and accuracy of some of the technical claims. The decision is to accept this paper, but I urge that authors to carefully address all reviewers' comments in the final version of this paper.

---

> ### Author Response · Authors · 2021-08-31
> **Response to Area Chair 4aGL**
>
> We thank AC for the helpful comments. We updated the manuscripts according to your and other reviewers' suggestions and responded to each reviewers’ questions and concerns. The following are responses for some of your concerns:
>
> 1. While we accepted and thanks for most suggestions of reviewers, **we do not agree with Reviewer ddzl’s some opinions about this work** including novelty, soundness, relevance with the CoRL, and value for the practical applications:
> * Novelty: To the best of our knowledge, our work is the first to consider the Egocentric and Allocentric view in the motion prediction problem. We believe this is an important insight.
> * Soundness: The proposed model has been compared with top-rank models on two popular datasets and achieved state-of-the-art performance. In Section 6, we compared the proposed model with several variants and verified its effectiveness. We also added experimental and implementation details in the Appendix D.
> * Relevance with the CoRL: Autonomous driving is a sub-area of robotics. Since our paper is learning-based trajectory prediction, it is robot learning. Actually, there are plenty of autonomous driving related papers accepted by CoRL 2020.
> * Value for the Practical Applications: the reviewer ddzl doubt the value of (i) trajectory prediction (ii) top-down view trajectory prediction for autonomous driving. We argue that it is a popular choice to do the prediction separately from perception and planning and to use the top-down view. In fact, lots of the works in the Related Works section are doing trajectory prediction in the top-down view.
>
> 2. As for https://arxiv.org/pdf/2007.12036, they studied the task of joint multi-agent prediction as well, which is highly relevant to our paper. However, it is difficult for us to conduct experiments since their method is based on LiDAR point clouds and the raster map which is different from the proposed 2D-coordinate-based model. The dataset they used are composed of LiDAR point clouds and maps. As suggested in their paper, other models need to use their perception backbone to do fair comparison. However, we could not find their code which makes it infeasible to compare on their dataset. As for the INTERACTION and TrajNet++, they are 2D-coordinate datasets without LiDAR point clouds, which makes it infeasible to compare on these two datasets. However, **we recognized the high relevance of this work in terms of joint multi-agent prediction and added some discussions to our Related Works section**.
>
> In conclusion, according to the reviewers’ suggestion, in the updated manuscripts, we
> 1.	adjusted the structure of the paper by moving Related Works to Section 2 and modifying the length and description of Section 6.
> 2.	added visualization of the proposed model against the baseline methods in the Appendix A to better show the effectiveness of the proposed method.
> 3.	added some discussions and experiments in the aspect of normalization about the proposed model in the Appendix B and C.
> 4.	added details about experiments, metrics, and implementations of the proposed model (Section 5.1, Appendix D).
> 5.	rewrote some parts suggested by the reviewers and corrected typos
>
> Thank you again for taking the time to review.

---

### Decision · Program_Chairs · 2021-09-13

**Decision:**

Accept (Poster)

**Comment:**

This paper proposes a novel method for multi-agent prediction. While the reviewers are generally positive about this paper, they also raise a number of important concerns (in particular reviewer ddzJ). The authors should carefully address the reviewers' comments in their rebuttal. Also, I would encourage the authors to compare their work with this paper: https://arxiv.org/pdf/2007.12036.pdf

UPDATE POST DISCUSSION PHASE: I would like to thank the authors for their comments and clarifications during the discussion phase. This paper was extensively discussed by the reviewers and the Area Chair. In general, there is agreement that this paper provides a useful contribution. Yet, concerns persist regarding a number of aspects of this paper, in particular regarding the rigor and accuracy of some of the technical claims. The decision is to accept this paper, but I urge that authors to carefully address all reviewers' comments in the final version of this paper.